# Cardiac Glycosides in Human Physiology and Disease: Update for Entomologists

**DOI:** 10.3390/insects10040102

**Published:** 2019-04-10

**Authors:** Rif S. El-Mallakh, Kanwarjeet S. Brar, Rajashekar Reddy Yeruva

**Affiliations:** Department of Psychiatry and Behavioral Sciences, University of Louisville School of Medicine, Louisville, KY 40202, USA; kjsb03@hotmail.com (K.S.B.); rajashekar.yeruva@louisville.edu (R.R.Y.)

**Keywords:** cardiac glycosides, cardenolides, sodium pump, Na,K-ATPase, ouabain, digoxin, digitoxin, entomology

## Abstract

Cardiac glycosides, cardenolides and bufadienolides, are elaborated by several plant or animal species to prevent grazing or predation. Entomologists have characterized several insect species that have evolved the ability to sequester these glycosides in their tissues to reduce their palatability and, thus, reduce predation. Cardiac glycosides are known to interact with the sodium- and potassium-activated adenosine triphosphatase, or sodium pump, through a specific receptor-binding site. Over the last couple of decades, and since entomologic studies, it has become clear that mammals synthesize endogenous cardenolides that closely resemble or are identical to compounds of plant origin and those sequestered by insects. The most important of these are ouabain-like compounds. These compounds are essential for the regulation of normal ionic physiology in mammals. Importantly, at physiologic picomolar or nanomolar concentrations, endogenous ouabain, a cardenolide, stimulates the sodium pump, activates second messengers, and may even function as a growth factor. This is in contrast to the pharmacologic or toxic micromolar or milimolar concentrations achieved after consumption of exogenous cardenolides (by consuming medications, plants, or insects), which inhibit the pump and result in either a desired medical outcome, or the toxic consequence of sodium pump inhibition.

## 1. Introduction

The manufacture of cardenolides and bufadienolides by disparate groups of plants and animals is believed to be an evolutionary strategy that serves to reduce or prevent damage inflicted by herbivores or predators. Multiple groups of insects have evolved various mechanisms to tolerate the glycosides and/or to take advantage of them by concentrating them in their tissues to make themselves less palatable to predators. These relationships have become iconic demonstrations of the key biologic processes of co-evolution [1], warning coloration [2], and Batesian and Müllerian co-mimicry [3,4], and are taught widely. Most entomologists are very aware of the role of these glycosides or cardenolides in insects, but are less aware of the expanding knowledge about their role in mammalian physiology and human disease. Knowledge of these aspects of cardenolides and bufadienolides enhances and enriches the understanding about the use of glycoside toxicity by insects to offset predation.

## 2. Cardenolides and Bufadienolides

Cardiac glycosides are large steroid-backboned compounds that have a wide variety of sources in nature. They bind to specific binding sites on the large, catalytic alpha subunit of the sodium- and potassium-activated adenosine triphosphatase (Na,K-ATPase), or sodium pump. The sodium pump has a ubiquitous distribution among nearly all eukaryotes and it plays a key role in maintaining an electrochemical gradient across cellular membranes. This gradient is required for a wide range of cellular processes such as maintenance of cellular volume, co-transport of extracellular nutrients and ions, and maintaining membrane excitability in some cells [5,6,7,8,9]. The role of the sodium pump is so vital, that its proper function is required for life [6]. Recent work has revealed many subtleties about the structure and function of the sodium pump that helps inform the biology of poisonous plants and the insects that feed on them.

## 3. Structure of the Sodium Pump

Nearly all eukaryotic organisms have a sodium pump (the only exception being yeast [10]). The sodium pump tends to be fairly well preserved across many groups such as fruit flies and humans [5,6].

The sodium pump is made up of 3 subunits. The alpha subunit is the largest protein, and has 10 transmembrane segments and five extracellular loops [11]. It has binding sites for sodium and potassium ions, as well as a magnesium ion cofactor. It also has a catalytic binding site for adenosine triphosphate. Importantly, it has an extracellular site for binding with glycosides. The affinity of the different subunits to cardiac glycosides is variable, being greatest for the alpha3 subunit; high but less for the alpha2 subunit; and lowest for the ubiquitous alpha1 subunit [12,13]. The distribution of the alpha3 subunit is predominantly neuronal [13]. Alpha2 is found in glial cells in the brain, as well as all muscle cells [13]. In humans, a fourth subunit is found in the testes [13].

Insects have only one alpha subunit [14,15,16]. Insects that feed on glycoside-containing plants frequently have single nucleotide mutations that allow them to be resistant to inhibition by the cardenolides [17,18].

The pump is anchored into the membrane with a smaller protein called the beta subunit. There are 3 isoforms of the beta subunit in mammals [19]. Any of the alpha isoforms can couple with any of the beta isoforms. A third subunit, FXYD, and also known as gamma, is one of the 7 FXYD identified [19,20].

## 4. Function of the Sodium Pump

The pump maintains an electrochemical gradient by pumping out 3 sodium ions out of the cell, in exchange for bringing in 2 potassium ions. This allows the intracellular sodium concentration to be quite low in comparison to the extracellular concentration, and the reverse for potassium. This requires the utilization of energy. In warm-blooded mammals, the sodium pump is the major source of non-shivering thermogenesis [21]. The sodium gradient is used by cells to cotransport glucose and amino acids [22], as well as control intracellular calcium concentrations [23], pH [23], volume [24], and membrane excitability in electrically active tissues [25]. Additionally, the pump will activate second messenger signals independent of any effect on sodium and potassium transport. Specifically, binding to the sodium pump activates epidermal growth factor receptors (EGFR) and the associated intracellular signaling that activates extracellular signal-regulated kinase (Erk) 1 and 2, phosphoinositide 3-kinase (PI3K) pathway, protein kinase B (also known as the serine/threonine kinase Akt), and protein 3-phosphoinositide dependent protein kinase-1 (PDK) [26,27,28]. Thus, ouabain can modulate both apoptosis and inflammation, and may have these effects at ouabain concentrations below those that inhibit ion transport [29,30].

## 5. Plant-Derived Cardenolides

A diverse range of plants have evolved the ability to synthesize or sequester cardiac glycosides. These are steroid backboned compounds that share a common steroid nucleus with 5-member lactone ring (cardenolides) or 6-member lactone ring (bufadienolides, occurring in toads, the Asian snake (*Rhabdophis tigrinus*) which sequesters them from prey toads, Lampyridae (fireflies) and some plants such as some Hyacinthaceae (subfamily Urgineoideae) such as the Egyptian squill, *Urginea maritima*) and contain a variety of combinations of hydroxyl, sulfate or carbohydrate groups [31,32,33]. In the 1970s, Skou was instrumental in identifying the sodium pump [34] and that these compounds directly interact with it [35]. He subsequently won the 1997 Chemistry Noble Prize for his work [34]. Cardenolides in plants do not appear to have intrinsic activity in the plant of origin, but appear to serve a protective function [36]. The most commonly recognized cardenolides are digoxin-derived from *Digitalis* spp. (foxglove) and utilized medically—and g-Strophanthin or ouabain—derived from *Strophantus gratus* (Apocynaceae) (climbing oleander) and found in *Acocanthera schimperi* (Apocynaceae) (Ouabaio tree) and used medically in the first half of the 20th century [37] and experimentally in the second half [38]. Both have quite high affinity to the sodium pump and will affect the heart at relatively low doses.

In entomology, the best-known glycoside-containing plants are the milkweeds (*Asclepias* spp., Asclepiadaceae). Eighteen of the 108 North American species contain glycosides [39,40]. These include calotropagenin, corotoxigenin, coroglaucigenin, syriogenin, uzarigenin, xysmalogenin, and many of their derivatives [41,42].

Medicinally, the Ebers Papyrus (1555 BC), an Egyptian medical papyrus, documents the use of plants containing cardioactive substances to treat cardiac alignments at least 3000 years ago (probably the Egyptian squill) [43]. William Withering, an English botanist and physician, was the first known physician to have used the extracts of foxglove plant (*Digitalis purpurea*) for the treatment of dropsy (edema) and other diseases [44]. Oswald Schmiedeberg, a German pharmacologist, isolated digitoxin in 1875, the active compound in the *Digitalis* extracts [44]. In traditional Chinese medicine, toad skins containing bufadienolides, and generally known as Chan Su, are also used medicinally for congestive heart failure and cancer [45]. Digitoxin and digoxin are still available for the treatment of congestive heart failure, but their use has dropped off after demonstrations that digoxin does not prolong life [46]. There are web sites that recommend that plant extracts containing cardiac glycosides can be used to commit suicide and homicide.

## 6. Sequestration of Cardenolides by Insects

A wide range of insects have co-evolved with glycoside-containing plants, so as to become able to sequester the glycosides within their tissues and protect them from predation. These insects have sodium pumps but circumvent toxicity by several mechanisms. Most have evolved a sodium pump that is resistant to the toxic effects of these compounds [18,47]. A common single amino acid substitution (N122H) in the alpha subunit of the sodium pump, confers cardenolide resistance to insects that are phylogenetically separated by 300 million years of evolution [18]. They are spread across 4 separate orders: *Danaus plexippus* (Lepidoptra, Nymphalidae, Danainae), *Liriomyza asclepiadis* (Diptera, Agromyzidae), *Labidomera clivicollis* (Coleoptera, Chrysomelidae); and *Oncopeltus fasciatus* and *Lygaeus kalmii* (Heteroptera, Lygaeidae) [18]. Other mechanisms include protective hemolymph. The oleander hawk-moth, *Daphnis nerii* (Sphingidae), does not have an insensitive form of the sodium pump but is still resistant to the effects of glycoside that are injected into its hemolymph. Instead, it possesses a unique perineurium barrier that prevents both polar and nonpolar glycosides from accessing neural tissues [48]; however, it is not clear how its muscle tissues are protected. The milk weed bug, *Oncopeltus fasciatus* (Heteroptera, Lygaeidae), will sequester the cardenolides within limited anatomical locations [49,50]. A few insects appear to also synthesize sufficient amounts of these compounds to also become toxic. These include Lampyridae and *Chrysolina* and *Chrysochloa* spp. (Chrysomelidae) [51,52].

The best-known examples of insects that sequester cardenolides are the Danain butterflies, including the Monarch, the Queen, and the Viceroy (the latter has also been found to be toxic, but less so than the Monarch it resembles [3]. These beautiful butterflies also attract lay attention because they migrate great distances and over winter as adults in both the United States and Mexico [53,54,55]. Initially, the apparent mimicry between the Monarch and the Viceroy was used as a text-book example of Batesian mimicry [3,4]. More recently, these butterflies have garnered attention for their declining numbers, a process believed to be caused by the disappearance of their food plant through the extensive use of herbicides in fields of genetically modified crops, the loss of their forest habitat in Mexico, and weather change [55,56].

## 7. Mammalian Endogenous Cardiotonic Steroids

Several researchers have independently confirmed the presence of endogenously synthesized ouabain-like, digoxin-like factors, or their dihydro-metabolites in mammals [57,58,59,60,61]. Bufadienolide-like factors have also been isolated [62,63,64]. Similar compounds have been identified in birds [65]. The most commonly studied of these is endogenous ouabain, which bears a high resemblance to the plant-derived form [65], and is widely present in many mammalian tissues [66]. It is also the endogenous cardenolide that has been most frequently associated with disease states (Table 1). Ouabain and other endogenous cardenolides are synthesized in the adrenal gland and the hypothalamus [67,68,69,70]. They are believed to be important in the regulation of the sodium pump, activation of cellular pathways [66], and may even serve as growth factors [66,71].

The body fluid concentrations of endogenous ouabain are quiet low (pM to nM range, [66,78]). At these physiologic picomolar to nanomolar concentrations ouabain actually increases sodium pump activity [71,78,85,86,87,88,89,90]. Higher, toxic or pharmacologic concentrations in the upper nanomolar and micromolar range are required for the inhibition of the pump [91,92]. This biphasic effect of ouabain and other cardenolides has been overshadowed by the experimental experience of so many scientists, and generally underappreciated [93]. Other actions such as the activation of intracellular and extracellular signaling pathways [93,94], and cellular growth or survival [66,71] also occur at this physiologic range. Despite the fact that the sodium pump is activated at lower concentrations of ouabain, intracellular calcium concentrations increase [95], which may mediate the pharmacologic effect of cardiac glycoside treatment on congestive heart failure. These actions are of particular importance since the activation of second messengers expands the effect of endogenous ouabain beyond the immediate actions on ion regulation. However, it raises the complication of determining how ouabain may have various actions while binding to the same receptor.

At higher (high nanomolar or micromolar) toxic concentrations, as may occur in a herbivore consuming a cardenolide-containing plant, or a bird eating a cardenolide-containing insect, the cardenolide will inhibit the pump activity [96]. This is associated with toxic effects on cells and cell survival [97,98].

In other words, the effect of ouabain on Na,K-ATPase activity and cellular function, is biphasic—initially stimulatory with antiapoptotic and cell growth effects at the lower physiologic concentrations, and then inhibitory with apoptosis and necrosis at the higher concentrations.

## 8. Physiologic Role of Endogenous Glycosides

Control of Na,K-ATPase activity is accomplished by a combination of: (1) concentrations of sodium or potassium ions across the membrane; (2) concentrations of the endogenous cardenolides; and (3) the isoform-specific tissue distribution of the pump. Excessive elaboration of endogenous cardiotonic steroids appears to be associated with some diseases such as congestive heart failure [72], renal dysfunction [99], or hypertension [74,75] (Table 1). Similarly, reductions in or deficiencies of endogenous cardenolides is associated with the abnormal growth of kidney cells [71,84] and mania in bipolar illness [8,78,100]. Nonetheless, it is not clear whether the changes of endogenous ouabain in these conditions are the cause or a consequence of the disease.

However, stressors that appear to require increased sodium pump activity are associated with the increased production of endogenous cardenolides and bufadienolides. These include conditions such as pregnancy [81] and exercise, which generally must be somewhat extreme to increase endogenous cardenolides [101], particularly ouabain [78,79]. Mice who have had the alpha2 subunit of the sodium pump knocked out, and consequently have reduced sodium pump activity [102], will also have increased endogenous ouabain production [103]. Thus, the production of endogenous ouabain appears, in part, to be linked to sodium pump activity. However, the list of conditions in which endogenous ouabain production appears to be increase (Table 1), suggests that other mechanisms, particularly those that might be linked to sodium regulation [104] may be involved.

Experimentally, cardiac glycosides are able to induce apoptosis in a variety of cell types [97,105,106] which has raised the possibility that these agents may have a particular anti-cancer effect [107]. However, the fact that the proapoptotic or necrotic effect is minimally selective for cancerous cells, and that at least some types of cancer are more resistant than normal tissue [108], reduces the enthusiasm for their practical utility.

## 9. Conclusions

A wide variety of plants elaborate cardenolide compounds as a defense against herbivory. Some insects have adapted to the presence of these compounds by developing a resistance to them and/or sequestering them, to utilize them as a defense against predation. Realizing that the action of cardenolides is biphasic is essential to understanding its physiologic and toxic roles. Mammals are known to synthesize endogenous cardenolides for their own physiologic needs. Lower, physiologic levels in the picomolar or lower nanomolar range generally increases pump activity, while higher, toxic levels, in the micromolar range, generally inhibit pump activity. Diseases have been associated with both an excess and a relative deficiency of endogenous cardenolides, particularly ouabain.

## Figures and Tables

**Table 1 insects-10-00102-t001:** Diseases or conditions associated with dysregulation of endogenous cardenolides.

Condition	Plasma Cardenolide Level	Fold Increase (Unless Noted to Be Reduction)	References
Congestive heart failure	CG: 23.3 ± 2.2 pg/mL		[72,73]
	CHF: 273.7 ± 35.5 pg/ mL	11.8×	
Essential Hypertension	CG: 76.3 ± 9.3 nM		[74,75]
	HTN: 234.8 ± 48.7 nM	3.0×	
Myocardial Infarction	CG: 0.04 ± 0.12 ng/mL		[62,76]
	MI: 1.65 ± 0.5 ng/mL	41.2×	
Supraventricular Tachycardia	CG: 29.4 ± 20.6 pM OE		[77]
	SVT: 35 + 18 pM OE	1.2×	
Exercise	Pre: 2.5 ± 0.5 nmol/L		[78,79]
	Post: 86 ± 27.2 nmol/L	34.4×	
Pregnancy	T 1: 16.3 ± 4.0 pmol/L	1.8×	[80]
	T 2: 18.8 ± 4.3 pmolL	2.0×	
	T 3: 24.3 ± 4.0 pmol/L	2.6×	
Pregnancy induced Hypertension	T3: 195 pg DE/mL		[81]
	PIH: 315 pg DE/mL	1.6×	
Pre-eclampsia	CG: 0.297 ± 0.037 nmol/L		[82]
	PEL: 0.697 ± 0.16 nmolL	2.3×	
Chronic Kidney Disease	CG: 24.7 ± 3.2 pg/mL		[75]
	CKD: 98.7 ± 17.4 pg/mL	4.0×	
Renal Cell Development *in utero*	Experimental model	80% reduction	[71]
Primary Hyperaldosteronism	CG: 1.06 ± 0.86 pM-OE/mL		[74]
	PHA: 2.59 ± 1.39 pM-OE/mL	2.4×	
Obstructive Sleep Apnea	CG: 110 ± 25 pM-OE/L		[83]
	OSA: 244 ± 51 pM-OE/L	2.2×	
Bipolar Illness	Reduced levels	0.5 reduction	[84]

CG: control group; CHF: congestive heart failure; HTN: hypertension; MI: myocardial infarction; SVT: supraventricular tachycardia; PIH: hypertension associated with pregnancy; PEL: pre-eclampsia; CKD: chronic kidney disease; PHA: primary hyperaldosteronism; OSA: obstructive sleep apnea.

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
