# Peer review of "Cardiac Glycosides in Human Physiology and Disease: Update for Entomologists"

_insects, 2019, doi:10.3390/insects10040102_

Round 1

Reviewer 1 Report

This manuscript summarizes some issues regarding the physiology and pharmacology of cardiac steroids to be published in literature that is followed by entomologists. There are several recent reviews on the subject (strangely, not cited by the authors) and I leave it to the editor to decide whether there is a place for such a summary specifically for Insects readers. There are topics that one would expect to be covered in more details in this review such as the involvement of cardiac steroids in inflammation and cancer.

Accuracy of references is always important and much more so in a review. The literature cited in this manuscript is poor, not updated and with numerous errors. It seems that the addition of the references was done by a post doctorate fellow who is new to the field.

The manuscript is poorly written and contains numerous errors,

Specific comments:

Line 10- Cardiac glycosides are cardenolides and bufadienolides – correct

Line 18- stimulation of the pump was shown only for ouabain (not digoxin)- correct

Line 27- see comment for Line 10

Line 38 – see comment for Line 10

Line 44 – these are old references replace with new reviews on the pump

Line 49 – “Nearly”? Do you know of eukaryotic cell that does not have sodium pump?

Lines 54-57 – strange and inaccurate statements- The affinity of cardenolides to different pump isoforms in rodents is different, while it is similar in human. Where did you see discrimination between binding and inhibition of activity? This is nonsense…

Line 59 – “prostate”? I guess you meant testis

Line 60 – insects have only one.. add “alpha subunit” and provide reference

Line 64- there are three isoforms of the betta subunit

Line 65- the third subunit (delete “final”) is FXYD, the gamma is one of the 7 FXYD identified.

Line 67- pumping out of the cell- add “out”

Line 73 – How can you write about sodium pump activity without addressing in details it’s signaling role?

Lines 75-80- long and not clear sentence- rephrase

Line 80- inappropriate references

Line 84- not accurate- ouabain was used clinically (see the many papers by Fuerstenwerth).

Line 94- dropsy is not congestive heart failure

Line 96- they are still being used… did you hear about Chansue?

Line 97- Digoxin is not Digitoxin and is being used all over the world

Line 106- add- “in the alpha subunit of the sodium pump”.

Line 129- why cite papers only from one group? There are at least 4 other groups that identified ouabain-like compounds.

Line 130- why cite papers only from one group? There are at least 4 other groups that identified bufadienolide-like compounds.

Lines 131-132 – this is not correct. There is a debate as to the identity of the endogenous ouabain (see Baecher S et al. Clin Chim Acta 2014).

Line 132-133 – this is not correct. Ouabain was associated to pathological states at least as bufadienolides. In any event, Table 1 summarizes only a small portion of the available literature. I don’t think that the information is of interest to entomologists and the Table should be deleted.

Lines 135- 137- adrenal gland- add “located”. Reference 50 does not show biosynthesis.

Line 136- what is “home organism’s own sodium pump?

Table 1 – As mentioned above my suggestion is to delete the table. This is a superficial summary and there are many important studies not cited here. define abbreviations CG, HTN, PIH… etc.

Line 142- references are inappropriate.

Line 143- reference 57 not used?

Lines 145-149- this must be developed and not presented as a side issue.

Line 146- reference 60 is not appropriate, use one of the reviews by Xie.

Line 147- the effect on calcium is through the effect on signaling.

Lines 152-153 – references are inappropriate.

Line 154-156 – this may be correct for effects on transport but what about the effects on signaling?

Line 163- very poor citations. Many important publications on kidney and bipolar illness were not cites.

Line 178 – one can understand from the text that the data on the effects of cardenolides on pump activity was done on the endogenous compounds - rephrase.

Author Response

This manuscript summarizes some issues regarding the physiology and pharmacology of cardiac steroids to be published in literature that is followed by entomologists. There are several recent reviews on the subject (strangely, not cited by the authors) and I leave it to the editor to decide whether there is a place for such a summary specifically for Insects readers. There are topics that one would expect to be covered in more details in this review such as the involvement of cardiac steroids in inflammation and cancer.

Accuracy of references is always important and much more so in a review. The literature cited in this manuscript is poor, not updated and with numerous errors. It seems that the addition of the references was done by a post doctorate fellow who is new to the field.

The manuscript is poorly written and contains numerous errors,

Specific comments:

1.            Line 10- Cardiac glycosides are cardenolides and bufadienolides – correct

Response:  The sentence was changed to read “Cardiac glycosides, cardenolides and bufadienolides, are elaborated by several plant or animal species to prevent grazing or predation.”

2.            Line 18- stimulation of the pump was shown only for ouabain (not digoxin)- correct

Response:  The sentence was changed to read “Importantly, at physiologic picomolar or nanomolar concentrations, endogenous ouabain, a cardenolide, stimulates the sodium pump. . .”

3.            Line 27- see comment for Line 10 (now line 28-29)

Response:  The sentence was changed to read “Manufacture of cardenolides and bufadienolides, by disparate groups of plants and animals is believed. . .”

4.  Line 38 – see comment for Line 10 

Response:  Subtitle changed to “2. Cardenolides and Bufadienolides

5.            Line 44 – these are old references replace with new reviews on the pump

Response:  While we do not think that an 8 year old reference is old, and we could not find more recent reviews that covered the breadth covered by the two papers we referenced.  Thus, instead of replacing these references we added newer ones that were focused:

Add:  Kay AR, Blaustein MP. Evolution of our understanding of cell volume regulation by the pump-leak mechanism.  J Gen Physiol. 2019 Feb 19. pii: jgp.201812274. doi: 10.1085/jgp.201812274.

Add: Lichtstein D, Ilani A, Rosen H, Horesh N, Singh SV, Buzaglo N, Hodes A. Na⁺, K⁺-ATPase Signaling and Bipolar Disorder. Int J Mol Sci. 2018 Aug 7;19(8). pii: E2314. doi: 10.3390/ijms19082314

Add:  Khalaf FK, Dube P, Mohamed A, Tian J, Malhotra D, Haller ST, Kennedy DJ. Cardiotonic Steroids and the Sodium Trade Balance: New Insights into Trade-Off Mechanisms Mediated by the Na/K-ATPase.  Int J Mol Sci 2018 Aug 30;19(9). pii: E2576. doi: 10.3390/ijms19092576.

6.  Line 49 (now 50-51) – “Nearly”? Do you know of eukaryotic cell that does not have sodium pump?

Response:  Yes, yeast are considered eukaryotes and do not have the Na,K-ATPase.  To clarify this we modified the sentence as follows: “Nearly all eukaryotic organisms have a sodium pump (the only exception being yeast [Farley et al., 1994]).”  (The added reference is Farley R.A., Eakle K.A., Scheiner-Bobis G., Wang K. (1994) Expression of Functional Na+/K+-ATPase in Yeast. In: Bamberg E., Schoner W. (eds) The Sodium Pump. Steinkopff DOIhttps://doi.org/10.1007/978-3-642- 72511-1_2)

7.  Lines 54-57 (now 57-58)– strange and inaccurate statements- The affinity of cardenolides to different pump isoforms in rodents is different, while it is similar in human. Where did you see discrimination between binding and inhibition of activity? This is nonsense… 

Response:  This was an error on our part.  We thank the reviewer for catching it.  The sentence has been modified as follows: “The affinity of the different subunits to cardenolides is the same [8] but the sensitivity of the change in activity to cardenolide binding is variable, being greatest for the alpha3 subunit; high . . .” 

8.  Line 59 (now 60) – “prostate”? I guess you meant testis

Response:  Yes. The reviewer is correct.  This sentence has been corrected:  “In humans, a fourth subunit is found in the testes [9].

9.  Line 60 (now 61) – insects have only one.. add “alpha subunit” and provide reference

Response: We added the word “alpha” and added a third reference to the two already there:  Emery AM, Billingsley PF, Ready PD, Djamgoz MBA Insect Na+/K+-ATPase. J Insect Physiol 1998; 44 (3-4): 197-210

10.  Line 64 (now 65) - there are three isoforms of the betta subunit

Response:  The reviewer is correct.  The sentence was changed to:  “.  There are 3 isoforms of the beta subunit in mammals [Clausen et al., 2017].  ”

Add reference:  Clausen MV, Hilbers F, Poulsen H.  The Structure and Function of the Na,K-ATPase Isoforms in Health and Disease. Front Physiol. 2017 Jun 6;8:371. doi: 10.3389/fphys.2017.00371.

11.  Line 65 (now 66-67) - the third subunit (delete “final”) is FXYD, the gamma is one of the 7 FXYD identified.

Response: The current sentence (“A final subunit, gamma, may modify ion transport of the pump [14].”) was replaced with the reviewer’s suggested wording above and the Clausen et al reference noted for line 64 comment was added, As follows:  “A third subunit, FXYD, and also known as gamma, is one of the 7 FXYD identified [14,Clausen]”

12.          Line 67 (now 69-70) - pumping out of the cell- add “out”

Response:  This was added, so that the sentence now reads: “The pump maintains an electrochemical gradient by pumping out 3 sodium ions out of the cell, in exchange for bringing in 2 potassium ions.”

13.          Line 73 (now 76-83) – How can you write about sodium pump activity without addressing in details it’s signaling role? 

Response: We added some information about the signaling actions of cardenolides as follows:  “Additionally, the pump will activate second messenger signals independent of any effect on sodium and potassium transport.  Specifically, binding to the sodium pump will epidermal growth factor receptor (EGFR) receptors and intracellular signaling that activates extracellular signal-regulated kinase (Erk) 1 and 2, phosphoinositide 3-kinase (PI3K) pathway, protein kinase B (also known as the serine/threonine kinase Akt), and protein 3-phosphoinositide- dependent protein kinase-1(PDK) [Haas et al., 2002; Yu et al., 2010; Wu et al., 2013].  Thus ouabain can modulate both apoptosis and inflammation, and may have these effects at ouabain concentrations below those that inhibit ion transport [Sibarov et l., 2012; Orellana et al., 2016]

Add:  Haas M, Wang H, Tian J, Xie Z. Src-mediated inter-receptor cross-talk between the Na+/K+-ATPase and the epidermal growth factor receptor relays the signal from ouabain to mitogen-activated protein kinases. J Biol Chem. 2002 May 24; 277(21):18694-702.

Add:  Yu HS, Kim SH, Park HG, Kim YS, Ahn YM.  Activation of Akt signaling in rat brain by intracerebroventricular injection of ouabain: a rat model for mania. Prog Neuropsychopharmacol Biol Psychiatry. 2010 Aug 16;34(6):888-94

Add:  Wu J, Akkuratov EE, Bai Y, Gaskill CM, Askari A, Liu L. Cell signaling associated with Na(+)/K(+)-ATPase: activation of phosphatidylinositide 3-kinase IA/Akt by ouabain is independent of Src. Biochemistry. 2013 Dec 17; 52(50):9059-67.

Add:  Sibarov DA, Bolshakov AE, Abushik PA, Krivoi II, Antonov SM. Na+,K+-ATPase functionally interacts with the plasma membrane Na+,Ca2+ exchanger to prevent Ca2+ overload and neuronal apoptosis in excitotoxic stress.  J Pharmacol Exp Ther. 2012 Dec; 343(3):596-607.

Add:  Orellana AM, Kinoshita PF, Leite JA, Kawamoto EM, Scavone C. Cardiotonic Steroids as Modulators of Neuroinflammation.  Front Endocrinol (Lausanne). 2016; 7():10

14.          Lines 75-80- long and not clear sentence- rephrase 

    Line 80 (now 88-90) - inappropriate references

Response:  We shortened and hopefully clarified and replaced the current references with more appropriate references as follows: “A diverse range of plants have evolved the ability to synthesize or sequester cardiac glycosides.  These are steroid backboned compounds that share a common steroid nucleus with 5-member lactone ring (cardenolides) or 6-member lactone ring (bufadienolides, occurring in toads, the Asian snake [Rhabdophis tigrinus] which sequesters them from prey toads, Lampyridae [fireflies] and some plants including some Hyacinthaceae [subfamily Urgineoideae] such as the Egyptian squill, Urginea maritima) and contain variety of combinations of hydroxyl, sulfate or carbohydrate groups [20-22].”

Omit current 20-22

Appropriate references replaced the original references:

Add:  Eisner T, Wiemer DF, Haynes LW, Meinwald J. Lucibufagins: Defensive steroids from the fireflies Photinus ignitus and P. marginellus (Coleoptera: Lampyridae).  Proc Natl Acad Sci U S A. 1978 Feb;75(2):905-8.

Add:  Kopp B, Krenn L, Draxler M, Hoyer A, Terkola R, Vallaster P, Robien W. Bufadienolides from Urginea maritima from Egypt. Phytochemistry. 1996 May;42(2):513-22.

Add:  Hutchinson DA, Mori A, Savitzky AH, Burghardt, GM, Wu X, Meinwald J, Schroeder FC. Dietary sequestration of defensive steroids in nuchal glands of the Asian snake Rhabdophis tigrinus.  Proc Natl Acad Sci U S A. 2007 Feb 13;104(7):2265-70.

15.          Line 84 - not accurate- ouabain was used clinically (see the many papers by Fuerstenwerth).

Response:  We were not aware of this literature, and thank the reviewer for pointing this out.  We corrected the paragraph: “The most commonly recognized cardenolides are digoxin – derived from Digitalis spp. (foxglove) and utilized medically – and g-Strophanthin or ouabain – derived from Strophantus gratus (Apocynaceae) (climbing oleander) and found in Acocanthera schimperi (Apocynaceae) (Ouabaio tree) and used medically in the first half of the 20th century [Fürstenwerth 2019] and experimentally in the second half [24].”

Add:  Fürstenwerth H. Comment on: Endogenous Ouabain and Related Genes in the Translation from Hypertension to Renal Diseases, Int. J. Mol. Sci. 2018, 19, 1948. Int J Mol Sci. 2019 Jan 24;20(3). pii: E505. doi: 10.3390/ijms20030505.

16.          Line 94 (now 107) - dropsy is not congestive heart failure

Response:  this was corrected

17.  Line 96 (now 108-110) - they are still being used… did you hear about Chansue?

Response:  This sentence was corrected:  “In traditional Chinese medicine, toad skins containing bufadienolides, and generally known as Chan Su,, are used medicinally also for congestive heart failure and cancer [31].”

18.          Line 97 (now 110-112) - Digoxin is not Digitoxin and is being used all over the world

Response:  The sentence was corrected:  “Digitoxin and digoxin are still available for treatment of congestive heart failure, but their use has dropped off after demonstration that digoxin does not prolong life [32].”

19.          Line 106 (n ow 120) - add- “in the alpha subunit of the sodium pump”.

Response:  This was added: “A common single amino acid substitution (N122H) in the alpha subunit of the sodium pump, confers cardenolide resistance . . .”

20.          Line 129- why cite papers only from one group? There are at least 4 other groups that identified ouabain-like compounds.

Line 130- why cite papers only from one group? There are at least 4 other groups that identified bufadienolide-like compounds.

Response:  We added two additional references for each compound from different researchers.

Add:  Qazzaz HM, Valdes R Jr.  Simultaneous isolation of endogenous digoxin-like immunoreactive factor, ouabain-like factor, and deglycosylated congeners from mammalian tissues. Arch Biochem Biophys. 1996 Apr 1;328(1):193-200.

Add:  Weinberg U, Dolev S, Werber MM, Shapiro MS, Shilo L, Shenkman L.  Identification and preliminary characterization of two human digitalis-like substances that are structurally related to digoxin and ouabain.  Biochem Biophys Res Commun. 1992 Nov 16;188(3):1024-9.

Add:  Lichtstein D, Kachalsky S, Deutsch J.  Identification of a ouabain-like compound in toad skin and plasma as a bufodienolide derivative.  Life Sci. 1986 Apr 7; 38(14):1261-70.

Add:  Yoshika M, Komiyama Y, Konishi M, Akizawa T, Kobayashi T, Date M, Kobatake S, Masuda M, Masaki H, Takahashi H.  Novel digitalis-like factor, marinobufotoxin, isolated from cultured Y-1 cells, and its hypertensive effect in rats.  Hypertension. 2007 Jan; 49(1):209-14.

21.          Lines 131-132 (now 146) – this is not correct. There is a debate as to the identity of the endogenous ouabain (see Baecher S et al. Clin Chim Acta 2014).

Response:  We change the sentence to read: “which is indistinguishable from bears high resemblance to the plant-derived form . . .”

21.          Line 132-133 (now 145) – this is not correct. Ouabain was associated to pathological states at least as bufadienolides. In any event, Table 1 summarizes only a small portion of the available literature. I don’t think that the information is of interest to entomologists and the Table should be deleted. 

Response:  Endogenous ouabain is certainly the most commonly studied endogenous sodium pump ligand.  Many of the associations with bufadeienolides have not been replicated.  Nonetheless, we changed the statement to read:  “Perhaps the most important The most commonly studied of these is endogenous ouabain . . .”  We disagree with the recommendation to omit the table.  We believe it is accurate, and if the entolmologic reader is not interested in the table, they can simply not read it.

22.          Lines 135-137 (now 153) - adrenal gland- add “located”. Reference 50 does not show biosynthesis. 

Response:  Reference 50 demonstrates CNS production.  We added 2 more references to demonstrate human specificity and hypothalamus production.

Add:  el-Masri MA, Clark BJ, Qazzaz HM, Valdes R Jr.  Human adrenal cells in culture produce both ouabain-like and dihydroouabain-like factors.  Clin Chem. 2002 Oct;48(10):1720-30.

Add:  Murrell JR, Randall JD, Rosoff J, Zhao JL, Jensen RV, Gullans SR, Haupert GT Jr. Endogenous ouabain: upregulation of steroidogenic genes in hypertensive hypothalamus but not adrenal. Circulation. 2005 Aug 30;112(9):1301-8. 

23.          Line 136- what is “home organism’s own sodium pump?

Response:  We simplified the sentence to read: “They are believed to be important in the regulation of the home organism’s own sodium pump, . . .”

24.          Table 1 – As mentioned above my suggestion is to delete the table. This is a superficial summary and there are many important studies not cited here. define abbreviations CG, HTN, PIH… etc.

Response:  We would like to not delete this table.  We have defined the abbreviations.

25.          Line 142 (now 160)- references are inappropriate.

Response:  That is an error.  We added three references by two groups that demonstrate that lower doses of ouabain stimulate the pump

Add:  Gao J, Wymore RS, Wang Y, Gaudette GR, Krukenkamp IB, Cohen IS, Mathias TR. Isoform-specific stimulation of cardiac Na/K pumps by nanomolar concentrations of glycosides. J Gen Physiol 2002;119:297–312.

Add:  Holthouser K, Mandal A, Merchant ML, Schelling JR, Delamere NA, Valdes R Jr., Tyagi SC, Lederer ED, Khundmiri SJ. Ouabain stimulates Na–K ATPase through sodium hydrogen exchanger-1 (NHE-1) dependent mechanism in human kidney proximal tubule cells. Am J Physiol Renal Physiol 2010; 299:F77–F90.

Add:  Khundmiri SJ, Salyer SA, Farmer B, Qipshidze-Kelm N, Murray RD, Clark BJ, Xie Z, Pressley TA, Lederer ED. Structural determinants for the ouabain-stimulated increase in Na–K ATPase activity. Biochimica et Biophysica Acta 2014;1843:1089–1102

26.          Line 143- reference 57 not used?

Response:  This has been deleted.

27.          Lines 145-149 (now 16—164) - this must be developed and not presented as a side issue.

Response:  The focus of the paper is to inform entomologists of the new world of discovery beyond effect on sodium pump.  We think that an extensive discussion of signal transduction is not appropriate for the audience of this paper.  However, we think highlighting the consequences of that is of interest.  We added the following:  “These actions are of particular importance since the activation of second messengers expands the effect of endogenous ouabain beyond the immediate actions on ion regulation.  However, it raises the complication of determining how ouabain may have various actions while binding to the same receptor.

Line 146 (now 160) - reference 60 is not appropriate, use one of the reviews by Xie.

Response:  We do not understand why the reviewer believes reference 60 is inappropriate, so we decided to keep it but added a review by Xie

Add:  Xie Z, Xie J. The Na/K-ATPase-mediated signal transduction as a target for new drug development. Front Biosci. 2005;10:3100-3109. 

28.          Line 147 (now 161-162) - the effect on calcium is through the effect on signaling.

Response:  This is a very good point because it is counter what one might expect.  We changed the sentence as follows: “Despite the fact that the sodium pump is activated at lower concentrations of ouabain, intracellular calcium concentrations increase [61], . . .” 

29.          Lines 152-153 – references are inappropriate.

Response:  We added 2 references.

Add: Chou WH, Liu KL, Shih YL, Chuang YY, Chou J, Lu HF, Jair HW, Lee MZ, Au MK, Chung JG. Ouabain Induces Apoptotic Cell Death Through Caspase- and Mitochondria-dependent Pathways in Human Osteosarcoma U-2 OS Cells. Anticancer Res. 2018 Jan;38(1):169-178.

Add:  Xiao AY, Wei L, Xia S, Rothman S, Yu SP.  Ionic mechanism of ouabain-induced concurrent apoptosis and necrosis in individual cultured cortical neurons.  J Neurosci. 2002 Feb 15;22(4):1350-62.

30.          Line 154-156 – this may be correct for effects on transport but what about the effects on signaling?

Response:  We changed the sentence to address this.  The effects on signaling are the same: “In other words, the effect of cardenolides on Na,K-ATPase activity and cellular function, is biphasic – initially stimulatory with antiapoptotic and cell growth effects at the lower physiologic concentrations, and then inhibitory with apoptosis and necrosis at the higher concentrations.”

31.          Line 163 (now 181-182) - very poor citations. Many important publications on kidney and bipolar illness were not cites.

The data regarding reduction of endogenous ouabain is more limited that excess.  There are many papers on excess, but this sentence is specifically about reduction.  Nonetheless, we added some more citations:

Add:  Manunta P, Messaggio E, Casamassima N, Gatti G, Carpini SD, Zagato L, Hamlyn JM. Endogenous ouabain in renal Na+ handling and related diseases. Biochim Biophys Acta 2010;1802(12):1214-1218. doi: 10.1016/j.bbadis.2010.03.001.

Add:  Lichtstein D, Ilani A, Rosen H, Horesh N, Singh SV, Buzaglo N, Hodes A.  Na, K-ATPase Signaling and Bipolar Disorder.  Int J Mol Sci. 2018 Aug 7;19(8). pii: E2314. doi: 10.3390/ijms19082314. 

32.          Line 178 – one can understand from the text that the data on the effects of cardenolides on pump activity was done on the endogenous compounds - rephrase.

Response: A lot of the data was derived from endogenous compound work (either measuring or blocking).  This is a review sentence.  No changes were made.

Reviewer 2 Report

This is an interesting review on the effects of low and high concentration of glycosides in human health. I am not so sure entomologist are interested in a review related to human disease on the topic of glycosides unless there is some relationship, such as human which might consume insects and then obtain glycosides in the diet. But maybe they would be degraded in digestion anyways.

Overall, this is a very good paper. It was very informative and synthesized a wide array of literature in demonstrating the functions of cardenolides across plants, insects, and mammals. I only have a few minor revisions/clarifications.

Specific points:

line 49: It might be good to mention different isoforms of the pump and varying affinities to ouabain.

So I am not sure this statement is full accurate on line 55 as different ages of cells have different binding affinities.

Two molecular forms of (Na+ + K+)-stimulated ATPase in brain. Separation, and difference in affinity for strophanthidin. Sweadner KJ. J Biol Chem. 1979 Jul 10;254(13):6060-7.

2. Line 90-100. It is surprising while discussing the history in the inhibitors of the pump that there is no mention of Skou, J. C. who obtained the Nobel Prize for understanding the function of the pump .

3. Line 112-114. Does it not strike the authors as odd to mention that the brain has some protective nature and shows insensitivity if injected into the hawk-moth since the heart, skeletal muscle and peripheral nerve terminals on muscles would be exposure to cardenolides when systemically injected into the open circulation of the insect?

4. line 134 “…above the kidneys” I don’t think defining the location is needed as well as “….in the brain”. To elementary to use

5. Line 152: a greater C2+ influx … could it not also be a lack of Ca2+ efflux ?

6. Line 171: This reads as one is speculating that there is a feedback mechanism in reduced pump activity that the body makes more ouabain. So what is the signal? After reading this review on glycosides I am struck by  trying to understand the mechanisms in the synthesis in mammals.  

7. Lines 55-56: “…but the sensitivity of change in activity of the different subunits to cardenolides is the same [8] but the sensitivity of the change in activity to cardenolide binding is variable…”

When the author later mentions that certain subunits have greater sensitivities to change in activity, does it mean that the pump with the sensitive subunit is more inhibited at high  cardenolide concentrations, more activated at low cardenolide concentrations (as the paper states with oubain), or both? Or does it mean that certain alpha subunits physically “deform” more in response to binding with the cardenolide?

8. Lines 63-64: “There are two isoforms of the beta subunit.”

Is this referring to mammals, insects, or both?

9. Table 1

This is very minor, but the table was somewhat difficult to read. The first column’s text is aligned with the center of the text in the second column. It may make it easier to read if the first line of each entry within the column aligned with each other like

Congestive Heart Failure      

CG: 23+/-…

11.8 x

[63,72]

CHF:………

Essential Hypertension       

CG:…

3.0 x

[64,65]

HTN:………

Or horizontal lines could be placed between each row to better distinguish which information goes with which condition.

Also, the column title “condition” should be capitalized

10. Section 8. Physiologic Role of Endogenous Cardenolides

Induction of apoptotic properties was briefly mentioned in the previous section, but the potential of cardenolides as an apoptosis inducing drug for diseases such as cancer could be worth mentioning in the conclusion as a potential real life application to continue further research if the author finds it appropriate.

L.A. Rascón-Valenzuela, C. Velázquez, A. Garibay-Escobar, W. Vilegas, L.A. Medina-Juárez, N. Gámez-Meza, R.E. Robles-Zepeda,

Apoptotic activities of cardenolide glycosides from Asclepias subulata,

Journal of Ethnopharmacology,

Volume 193,

2016,

Pages 303-311,

This paper demonstrates that certain cardenolides induced cell death through caspase-dependent apoptosis. They tested these compounds on different cancerous cells and measured several markers of apoptosis such as caspase activity and mitochondrial membrance depolarization.  

Bloise, E., Braca, A., De Tommasi, N. et al. Cancer Chemother Pharmacol (2009) 64: 793. https://doi.org/10.1007/s00280-009-0929-5

This paper also tested the effects of cardenolides in cancerous cells and used cytofluorimetry and Western Blotting to assess the presence of key chemical markers of apoptosis. They found a reduction in cell number due to activation of caspace-dependent apoptotic pathways. They also found significant cell impairment in U937 cells, which did not experience as much cell death compared to other more susceptible cancer cells. This paper aimed to demonstrate the potential that cardenolides might have as an anticancer drug.

Author Response

This is an interesting review on the effects of low and high concentration of glycosides in human health. I am not so sure entomologist are interested in a review related to human disease on the topic of glycosides unless there is some relationship, such as human which might consume insects and then obtain glycosides in the diet. But maybe they would be degraded in digestion anyways.

Overall, this is a very good paper. It was very informative and synthesized a wide array of literature in demonstrating the functions of cardenolides across plants, insects, and mammals. I only have a few minor revisions/clarifications. 

Specific points:

1.            line 49 (now 51-52 and 57-59): It might be good to mention different isoforms of the pump and varying affinities to ouabain.  So I am not sure this statement is full accurate on line 55 as different ages of cells have different binding affinities.

Two molecular forms of (Na+ + K+)-stimulated ATPase in brain. Separation, and difference in affinity for strophanthidin. Sweadner KJ. J Biol Chem. 1979 Jul 10;254(13):6060-7.

Response:  This is included in the next sentence and paragraph.  Additionally, this was addressed in Reviewer 1’s comment number 7 above. As noted in that comment, there was an error in how the sentence was written that resulted in it providing a statement that was not accurate.  This has now been corrected.

2. Line 90-100 (now 92-94). It is surprising while discussing the history in the inhibitors of the pump that there is no mention of Skou, J. C. who obtained the Nobel Prize for understanding the function of the pump . 

Response:  This is a very nice point.  This was added:  “In the 1970s Skou was instrumental in identifying the sodium pump {Skou 2004] and that these compounds directly interact with the it [Hansen and Skou 1973].  He subsequently won the 1997 Chemistry Noble Prize for his work [Skou 2004]..

Add:  Skou JC. The identification of the sodium pump.  Biosci Rep 2004 Aug-Oct;24(4-5):436-51.

Add:  Hansen O, Skou JC.  A study on the influence of the concentration of Mg2+, Pi, K+, Na+, and Tris on (Mg2+ + Pi)-supported g-strophanthin binding to (Na+ = K+)-activated ATPase from ox brain.  Biochim Biophys Acta. 1973 Jun 7;311(1):51-66.

3. Line 112-114. Does it not strike the authors as odd to mention that the brain has some protective nature and shows insensitivity if injected into the hawk-moth since the heart, skeletal muscle and peripheral nerve terminals on muscles would be exposure to cardenolides when systemically injected into the open circulation of the insect?

Response:  This is a good point, but we could not find any literature that addresses this.  We could only change the discussion to the following: “Rather, it possesses a unique perineurium barrier that prevents both polar and nonpolar glycosides from accessing neural tissues [34]; however, it is not clear how muscle tissues are protected.

4. line 134 (now 152-153) “…above the kidneys” I don’t think defining the location is needed as well as “….in the brain”. To elementary to use

Response:  The sentence was simplified as follows:  “Ouabain and other endogenous cardenolides are synthesized in the adrenal gland and the hypothalamus [49,50].

5. Line 152 (now 173-174): a greater C2+ influx … could it not also be a lack of Ca2+ efflux ?

Response:  The reviewer is correct.  However, the mechanism is not fully clear.  This sentence was also changed to address question 29 of reviewer 1 (see 29 above) and now reads as follows:  “This is associated with toxic effects on cells and cell survival.

6. Line 171 (now 192-195): This reads as one is speculating that there is a feedback mechanism in reduced pump activity that the body makes more ouabain. So what is the signal? After reading this review on glycosides I am struck by trying to understand the mechanisms in the synthesis in mammals.  

Response: The reviewer is correct, this is speculation, but it is something our laboratory has been focused on for a while.  We are currently preparing a paper that discusses the role of atrial naturetic peptide, but it is not yet peer-reviewed.  However, we added a theoretical paper from our laboratory that is peer-reviewed and changed the discussion as follows:  “Thus, production of endogenous ouabain appears, in part, to be linked to sodium pump activity. However, the list of conditions in which endogenous ouabain production appears to be increase (Table 1), suggests that other mechanisms, particularly those that might be linked to sodium regulation [Brar et al., 2016] may be involved.

Add:  Brar KS, Gao Y, El-Mallakh RS.  Are endogenous cardenolides controlled by Atrial Natriuretic Peptide?  Med Hypotheses 2016;92:21-25, 2016

7. Lines 55-56 (now 57-58): “…but the sensitivity of change in activity of the different subunits to cardenolides is the same [8] but the sensitivity of the change in activity to cardenolide binding is variable…”

When the author later mentions that certain subunits have greater sensitivities to change in activity, does it mean that the pump with the sensitive subunit is more inhibited at high cardenolide concentrations, more activated at low cardenolide concentrations (as the paper states with ouabain), or both? Or does it mean that certain alpha subunits physically “deform” more in response to binding with the cardenolide? 

Response: This sentence was corrected in response to reviewer 1 item 7 above.  However, the points brought up by this reviewer have been one of the major reasons that some scientists are not willing to accept the biphasic curve of ouabain on the sodium pump.  It is not clear how binding to the same receptor can both stimulate and inhibit sodium pump activity.  Activation of second messengers may provide the answer – activation of signaling pathways may stimulate the pump, but then as the inhibitory effect of binding kicks in, there is pump inhibition.  This is supported by various experimental data, but remains speculation and so was not included in the paper.

8. Lines 63-64 (now 65-66): “There are two isoforms of the beta subunit.”  Is this referring to mammals, insects, or both?

Response:  We are unable to find any reference that describes more than one beta subunit in insects.  The sentence was also changed in response to reviewer 1’s comment  above:

9. Table 1

This is very minor, but the table was somewhat difficult to read. The first column’s text is aligned with the center of the text in the second column. It may make it easier to read if the first line of each entry within the column aligned with each other like                                                   

Congestive Heart   Failure       

CG: 23+/-…

11.8 x

[63,72]

CHF:………

Essential Hypertension        

CG:…

3.0 x

[64,65]

HTN:………

Or horizontal lines could be placed between each row to better distinguish which information goes with which condition. 

Also, the column title “condition” should be capitalized 

Response:  These changes are for the typesetting folks. 

10. Section 8. Physiologic Role of Endogenous Cardenolides

Induction of apoptotic properties was briefly mentioned in the previous section, but the potential of cardenolides as an apoptosis inducing drug for diseases such as cancer could be worth mentioning in the conclusion as a potential real-life application to continue further research if the author finds it appropriate. 

L.A. Rascón-Valenzuela, C. Velázquez, A. Garibay-Escobar, W. Vilegas, L.A. Medina-Juárez, N. Gámez-Meza, R.E. Robles-Zepeda,

Apoptotic activities of cardenolide glycosides from Asclepias subulata,

Journal of Ethnopharmacology,

Volume 193,

2016,

Pages 303-311,

This paper demonstrates that certain cardenolides induced cell death through caspase-dependent apoptosis. They tested these compounds on different cancerous cells and measured several markers of apoptosis such as caspase activity and mitochondrial membrance depolarization.   

Bloise E, Braca A, De Tommasi N, Belisario MA. Pro-apoptotic and cytostatic activity of naturally occurring cardenolides. Cancer Chemother Pharmacol (2009) 64: 793. https://doi.org/10.1007/s00280-009-0929-5

This paper also tested the effects of cardenolides in cancerous cells and used cytofluorimetry and Western Blotting to assess the presence of key chemical markers of apoptosis. They found a reduction in cell number due to activation of caspace-dependent apoptotic pathways. They also found significant cell impairment in U937 cells, which did not experience as much cell death compared to other more susceptible cancer cells. This paper aimed to demonstrate the potential that cardenolides might have as an anticancer drug.

Response:  We chose to specifically use the 2 references provided by the reviewer in addition to a couple of others.  However, this does not have any clinical application since this effect is true of noncancerous cells as well, and cancerous cells may actually be more resistant to ouabain-induced apoptosis.  A paragraph was added at the end of this section: “Experimentally, cardiac glycosides are able to induce apoptosis in a variety of cell types [e.g., Bloise et al., 2009; Rascón-Valenzuela et al. 2016; Chou et al., 2018] which has raised the possibility that these agents may have a particular anti-cancer effect [Chen et al., 2006].  However, the fact that the proapoptotic or necrotic effect is minimally selective for cancerous cells, and that at least some types of cancer are more resistant than normal tissue [Clifford and Kaplan 2013], reduces the enthusiasm of their practical utility.

Add:  Bloise E, Braca A, De Tommasi N, Belisario MA. Pro-apoptotic and cytostatic activity of naturally occurring cardenolides. Cancer Chemother Pharmacol (2009) 64: 793. https://doi.org/10.1007/s00280-009-0929-5

Add:  Rascón-Valenzuela LA, Velázquez C, Garibay-Escobar A, Vilegas W, Medina-Juárez LA, Gámez-Meza N, Robles-Zepeda RE. Apoptotic activities of cardenolide glycosides from Asclepias subulata.  J Ethnopharmacol 2016; 193:303-311.

Add: Chou WH, Liu KL, Shih YL, Chuang YY, Chou J, Lu HF, Jair HW, Lee MZ, Au MK, Chung JG. Ouabain Induces Apoptotic Cell Death Through Caspase- and Mitochondria-dependent Pathways in Human Osteosarcoma U-2 OS Cells. Anticancer Res. 2018 Jan;38(1):169-178.

Add:  Chen JQ, Contreras RG, Wang R, Fernandez SV, Shoshani L, Russo IH, Cereijido M, Russo J.  Sodium/potassium ATPase (Na+, K+-ATPase) and ouabain/related cardiac glycosides: A new paradigm for development of anti- breast cancer drugs?  Breast Cancer Res Treat. 2006 Mar;96(1):1-15.

Add:  Clifford RJ, Kaplan JH.  Human breast tumor cells are more resistant to cardiac glycoside toxicity than non-tumorigenic breast cells.  PLoS One. 2013 Dec 13;8(12):e84306. doi: 10.1371/journal.pone.0084306.

Round 2

Reviewer 2 Report

The authors addressed all my concerns.